# Effect of Contextualized Versus Non-Contextualized Interventions for Improving Hand Washing, Sanitation, and Health in Rural Tanzania: Study Design of a Cluster Randomized Controlled Trial

**DOI:** 10.3390/ijerph16142529

**Published:** 2019-07-15

**Authors:** Kim Dockx, Hans Van Remoortel, Emmy De Buck, Charlotte Schelstraete, An Vanderheyden, Tiene Lievens, John Thomas Kinyagu, Simon Mamuya, Philippe Vandekerckhove

**Affiliations:** 1Centre for Evidence-Based Practice (CEBaP), Belgian Red Cross, Mechelen 2800, Belgium; 2Faculty of Medicine, Department of Public Health and Primary Care, KU Leuven, Leuven 3000, Belgium; 3Belgian Red Cross, Mechelen 2800, Belgium; 4Tanzania Red Cross Society, Dar es Salaam, Tanzania; 5Department of Environmental and Occupational Health, Muhimbili University of Health and Allied Sciences, Dar es Salaam, Tanzania; 6Faculty of Medicine, Department of Public Health and Primary Care, KU Leuven, Leuven 3000, Belgium

**Keywords:** Hand-washing, sanitation, RANAS, diarrhea

## Abstract

Nearly 90% of diarrhea-related mortalities are the result of unsafe drinking water, poor sanitation, and insufficient hygiene. Although “Water, Sanitation, and Hygiene” (WASH) interventions may significantly reduce the risk of diarrheal disease, it is currently unclear which interventions are the most effective. In this study, we aim to determine the importance of contextualizing a WASH intervention to the local context and the needs for increasing impact (Clinicaltrials.gov NCT03709368). A total of 1500 households in rural Tanzania will participate in this cluster randomized controlled trial. Households will be randomized into one of three cohorts: (1) a control group receiving a basic intervention and 1 placebo household visit, (2) an intervention group receiving a basic intervention + 9 additional household visits which are contextualized to the setting using the RANAS approach, and (3) an intervention group receiving a basic intervention + 9 additional household visits, which are not contextualized, i.e., a general intervention. Assessments will take place at a baseline, 1 and 2 years after the start of the intervention, and 1 year after the completion of the intervention. Measurements involve questionnaires and spot checks. The primary outcome is hand-washing behavior, secondary objectives include, the impact on latrine use, health, WASH infrastructure, quality of life, and cost-effectiveness.

## 1. Introduction

Diarrhea is ubiquitous among people in low-and middle-income countries (LMIC) [1]. Each year, approximately 1.7 billion children globally are faced with diarrheal disease. Children under five years of age are particularly vulnerable, with 525,000 mortalities per year [2]. Unsafe drinking water, poor sanitation, and insufficient hygiene are responsible for nearly 90% of these mortalities [3,4,5].

“Water, Sanitation, and Hygiene” (WASH) interventions were demonstrated to significantly reduce the risk of diarrheal disease [6]. Hand-washing with water and soap, in particular, is shown to significantly reduce the microbial load of the hands [7] and was shown to reduce the risk of diarrhea with 39–47% [6,7,8,9,10]. Despite its enormous health impacts, only 5–25% of people in LMIC are estimated to wash their hands with water and soap after fecal contact [10,11]. In a Cochrane systematic review by Ejemot-Nwadiaro et al., attempts to change either hygiene practices [12], hand-washing at critical times [13], or soap consumption [14] only led to modest improvements [9].

Several studies have suggested that multifaceted interventions, including both hardware (i.e., improvement of infrastructure) and software (i.e., improvement of knowledge, skills, and attitude), are needed to obtain lasting behavioral changes [15,16,17]. In addition, contextualizing the interventions and ensuring that the program is tailored to the needs of the participants may further increase impact [16]. Despite the many reports that have been published in recent years, a recent mixed methods systematic review by De Buck et al. showed that there is no consensus as to the most efficacious approach in improving WASH behavior in LMIC [18]. The systematic review also showed that contextualized hand-washing interventions using the ‘Risks, Attitudes, Norms, Abilities, Self-regulation’ (RANAS) model— a theoretical framework which can be used to tailor the content of the intervention to the context at play—show great promise for ameliorating behavior [18]. Indeed, RANAS trials were published to be successful in improving water [19,20,21], sanitation [22,23], and hygiene [24,25,26] behavior in various settings around the world. However, these trials made use of a low-quality, uncontrolled before-after study design, and largely focused on short-term behavior change, while long-term changes have gone uninvestigated. Moreover, the effects on health tend to be insignificant. With this study, we want to determine whether multifaceted contextualized interventions, based on the RANAS model versus non-contextualized WASH interventions have a differential impact on behavior and related health outcomes. The evidence regarding the efficacy of WASH interventions is not ideal, particularly as large RCTs are needed, which do not assist in determining long-term effects.

The primary objective of this study is to determine the effectiveness of add-on contextualized and non-contextualized interventions on hand-washing behavior at critical times. As a secondary objective, the study aims to evaluate the impact on latrine use, health, quality of life and hardware coverage. It will also calculate the cost-effectiveness of each approach.

## 2. Materials and Methods

To draft the protocol, we made use of the reporting criteria provided in the SPIRIT checklist (Appendix A).

### 2.1. Design

This study is a prospective, parallel group, single blinded, cRCT with a 2 year implementation period and a 12 month follow-up. It is a collaboration between Belgian Red Cross (BRC), Tanzania Red Cross Society (TRCS), and the Environmental and Occupational Health Department of Muhimbili University of Health and Allied Sciences (MUHAS).

Participants will be randomized to one of three cohorts: A contextualized intervention cohort, a non-contextualized intervention cohort, or a control cohort. A contextualized intervention is defined as an intervention that is adapted to the local context by collecting data at baseline and using this data to fit the intervention to the specific population needs, as prescribed by the RANAS approach. In contrast, a non-contextualized intervention is a general WASH intervention that is not fine-tuned to meet the specific needs of the context at hand. For the purpose of this study, the target population will be subdivided into three cohorts: (1) a control cohort receiving a basic intervention + 1 placebo household visit, (2) a contextualized intervention cohort receiving a basic intervention + 9 household visits which are contextualized to the setting using the RANAS approach, and (3) a non-contextualized intervention cohort receiving a basic intervention + 9 household visits which are not contextualized, i.e., a general intervention. The study flow chart can be found in Figure 1.

### 2.2. Participants and Setting

Participants will be recruited from seven villages in Buhigwe district, Kigoma region, Tanzania. Formative research showed that an estimated 4782 households live in this area, of which a random sample of 1500 households will be included in the study. Sampling was done by an independent researcher (KD) who was not involved in data collection or implementation of the interventions. A household is defined as one or more people who occupy a housing unit. Subjects were excluded as a respondent to the questionnaires if they were below the age of 18 years old at the time of the study. There were no other eligibility criteria.

### 2.3. Sample Size Calculation

The primary outcome of the study is the prevalence of hand-washing after defecation or latrine using. This prevalence will be compared between intervention and control groups at post-intervention. Since we have an interest in comparing both interventions groups separately, the alpha-level will be set at 0.025 (0.05/2).

The sample size estimate was based on the literature and earlier pilot work, which assumed that 5% of the households wash their hands after defecation, at a baseline, and a 15% improvement is expected following contextualized and non-contextualized interventions. To detect an increase in hand-washing prevalence after defecation or latrine using 5% to 20%, 92 households are needed per group to have 80% power, based on a two-sided Chi-square test (α = 0.025). However, the drop-out rate and intra-cluster correlation (ICC) still need to be taken into account. The ICC refers here to the potential correlation in the probability of hand-washing between the households of the same sub-village. The inflation in sample size due to the ICC is given by 1 + ICC*(*m −* 1), where *m* equals the mean number of households per sub-village. 

With 500 households per group (clustered in 9 sub-villages), including a drop-out rate of 20%, the variance inflation factor should be maximal 4.35 (400/92), corresponding to an ICC of 0.076 (4.35 = 1 + 0.076 × (45 − 1)). This closely corresponds to the most conservative estimate found in literature for a comparable outcome [27,28,29]. The calculation is also conservative, since the analysis will be based on a longitudinal statistical model, including data from earlier time points for drop-out participants and the baseline measurement will be used as a covariate. These further increases the power of the study. In conclusion, a total sample of 1500 households were included, i.e., 500 households per cohort. A detailed overview can be found in Table 1.

### 2.4. Allocation and Sampling

In a first step, formative research was done to create a list of all households living in the area. A household number was assigned to each of these households (*n* = 4782). Cluster randomization was used to assign households to one of the three cohorts. Households from the same sub-village will all be assigned to the same treatment arm. The sub-villages will be stratified according to size and whether or not the sub-village has a school. A total of four schools are available in the area. This implies that each cohort will contain at least one school, and one cohort will contain two schools. A detailed overview of the sub-villages with, and without, a school can be found in Table 1. The randomization of the sub-villages to one of the three cohorts will be done by an independent researcher (KD), using a computer-generated random list with R studio (Version 1.0.143, RStudio Inc. Boston, MA, USA) [30].

Following allocation, a representative sample was selected in each sub-village. This sample received the add-on interventions and assessments. As determined by the sample size calculation, the sample should include 31% of the total population living in the intervention area—i.e., 1500 households were included from a total of 4782 households living in the area. As such, 31% of the households were randomly selected using a computer-generated random list with R studio (Version 1.0.143). This sample was proportionally selected to the size of the sub-village (Table 1).

### 2.5. Description of the Interventions

The intervention was subdivided into a basic intervention, which provides all cohorts equally, and add-on household visits, which differ from one cohort to another. All interventions will be delivered by TRCS volunteers, who received formal training prior to implementation. Participants and TRCS volunteers were not blinded for the intervention status. 

#### 2.5.1. Basic Intervention

The basic intervention involves both hardware and software components. Hardware interventions are aimed at improving the infrastructure: An existing water gravity flow scheme will be rehabilitated and extended, 350 tippy taps and 350 pans for pour flush latrines will be distributed, and sanitation blocks will be built in four schools. In addition, community sessions will be provided to each of the sub-villages (*n* = 27), teaching participants how to build a tippy tap, how to build an improved latrine, and how to make liquid soap.

Software interventions involve the improvement of knowledge, skills, and attitude. In order to reach this goal, Community-Led Total Sanitation (CLTS), Participatory Hygiene and Sanitation Transformation (PHAST), and School Water, Sanitation and Hygiene (SWASH) clubs will be used. CLTS sanitation and hand-washing sessions will follow the National Guidelines for Rural Community Led Total Sanitation (R-CLTS) [31], and UNICEF material [32]. In brief, a 1 h meeting will take place in each of the villages involving community leaders and locals to familiarize them with the intervention, and to ensure optimal co-operation during implementation. Then, two community sessions with a duration of 3–5 h each will be organized in each of the sub-villages separately (*n* = 27). A first session will focus on sanitation [31], and a second session will focus on hand-washing [32]. Finally, a series of follow-up community meetings and household visits will take place to monitor improvement. PHAST sessions are based on a manual from the World Health Organization [33]. Considering the overlap with some CLTS activities, only a selection of the full guideline will be enrolled, namely: (1) Health problems in our community, (2) Good and bad hygiene behaviors, (3) Investigating community practices, (4) How diseases spread, (5) Blocking the spread of disease, (6) Selecting the barriers, (7) Choosing sanitation improvements, and (8) Choosing improved hygiene behaviors. These 8 activities will be enrolled in approximately 40 community groups, which are spread out over the sub-villages. Each activity will take between 30 min–2 h to complete.

#### 2.5.2. Add-On Intervention

All cohorts will receive an add-on intervention on top of the basic intervention package. The content of this add-on intervention will differ from one cohort to another.

##### Contextualized Intervention Cohort

The contextualized intervention cohort will receive nine household visits of 20–40 min each, at a frequency of one visit every two months. The content of these add-on visits is based on the ‘Risks, Attitudes, Norms, Abilities, Self-regulation’ (RANAS) model (www.ranasmosler.com) [16]. The RANAS model consists of five behavioral factors, which are considered to be the drivers of WASH behavior [16] (see Figure 2). In a preparatory phase, RANAS dictates that questionnaires and interviews are used to better understand the context at play, i.e., do people in the intervention area wash their hands and why (not)? In a second phase, this data is used to develop the intervention, thus optimally tailoring the program to the needs in the community [16,34].

Concretely, baseline data are first gathered through a questionnaire (Appendix A). Then, these data will be analyzed to compare (1) people who always wash hands after defecation (do’ers) to people who don’t (non-do’ers); and (2) people who always use the latrine (do’ers) to people who do not (non-do’ers). Do’ers were defined as people who always wash their hands or use the latrine. Non-do’ers were defined as people who wash their hands or use the latrine most of the time, sometimes, seldom or never. Only those behavioral factors (i.e., ‘Risks, Attitudes, Norms, Abilities, and/or Self-Regulation’), that are significantly different between “do’ers” and “non-do’ers” will be included in the intervention, as these are, at least theoretically, considered to be decisive for hand-washing and latrine use behavior in the current setting [16,34]. This implies that the precise content of the intervention can only be determined after baseline data collection has taken place.

##### Non-Contextualized Intervention

The non-contextualized intervention cohort will also receive nine household visits of 20–40 min each, at a frequency of one visit every two months. The content of these visits involves a general WASH intervention which is not tailored to the context at hand. Based on a study by Mosler et al., who suggested that norms are a universal driver (i.e., not context specific) of hand-washing behavior [35] and other studies and expert input stating that norms of local leaders, in particular, are important in African culture in inducing behavioral change [36,37,38], the focus would be non-contextualized intervention on norms. Concretely, the ‘Norm Behavior Change Techniques’ from the RANAS manual (www.ranasmosler.com) were used to outline the intervention [16]. These interventions encourage participants to improve their hand-washing and sanitation infrastructure and behavior in order to become a WASH role model within their community. A detailed outline of the non-contextualized intervention can be found in Appendix A.

##### Control

The control cohort will receive one household visit of 20–40 min. During this visit, a placebo poster will be distributed focusing on malaria nets. The main goal of this household visit is to avoid jealousy among the control cohort.

### 2.6. Assessment Protocol

A repeated measures design will be employed with assessments performed at baseline (May 2018), after one year (May 2019) and after two years (May 2020) of intervention, and at 1 year of follow-up, i.e., one year after the intervention has stopped (May 2021) (Figure 2). All assessments will be performed by trained assessors who are not involved in the implementation of the interventions, and who are blinded to group allocation. Assessments will take place at about the same time of year to avoid variability of performance due to the rainy season.

All data will be collected by means of KoboToolbox (https://www.kobotoolbox.org/), or on paper (EQ-5D-3L + informed consent). Moreover, collected data are checked daily during data collection by means of a computer script (R studio), to ensure that no mistakes have taken place – i.e., doubles in the dataset.

### 2.7. Outcome Measures

#### 2.7.1. Primary Outcome

**Hand-washing behavior.** The primary outcome is the percentage of households washing hands. The outcome measure will be obtained by means of a self-developed WASH questionnaire, which involves both multiple choice questions and spot checks (Appendix A). 

Spot checks will be used to determine the hardware situation in the household. Assessors will check whether there is a hand-washing station available, and if so, what type of hand-washing station (i.e., running water or a bucket). Spot checks will also be used to determine the presence of water and/or soap at the time of the measurement. By means of the questionnaire, assessors will probe when people normally wash their hands (i.e., before eating, after using the latrine, etc.), and what they use for hand-washing (i.e., water and/or soap). 

A household was defined to have correct hand-washing behavior when they: (1) have a hand-washing station available (spot check), (2) have water and soap available at the hand-washing station (spot check), and (3) indicate that they wash their hands with water and soap (questionnaire). All items must be present. The analysis will be repeated for each of the critical times. Critical times of interest include: after defecation or using the latrine, before cooking or handling food, before eating, and before feeding a child.

#### 2.7.2. Secondary Outcomes

**Latrine use.** Latrine use is assessed by means of a self-developed WASH questionnaire, involving both multiple choice questions and spot checks (Appendix A).

Spot checks will be used to check whether there is a latrine available at the household, and if so, what the latrine looks like (i.e., does it have a roof, a door, etc.), and to determine the latrine cleanliness. By means of the questionnaire, behavioral information about the past two weeks is gathered. These questions probe whether people (1) normally use a latrine when defecating at home; (2) normally use a latrine when defecating elsewhere (i.e., when at work in the field); (3) sometimes defecate in the open when at home; (4) sometimes defecate in the open when they are elsewhere; (5) clean the latrine and how often.

A household is defined to have correct latrine use behavior when: (1) they have a latrine available (spot check), (2) the latrine minimally includes walls, a roof, a door or curtain, and a slab or concrete floor (spot check), (3) the latrine is clean (spot check), (4) they indicate to normally use the latrine when defecating at home (questionnaire), and (5) they indicate to not defecate in the open when they are at home (questionnaire). A similar analysis will be performed for latrine use behavior when people are elsewhere.

**Health.** Health is assessed using a self-developed health questionnaire (Appendix A). The questionnaire is used to probe the prevalence rates of diarrhea, vomiting, limitations of daily activities, the need for medical care due to diarrheal illness in the past two weeks, and hospitalization due to diarrheal illness in the past three months. The WHO definition will be used: ‘Diarrhea is defined as the passage of three or more loose or liquid stools per day or more frequent passage than is normal for the individual. Frequent passing of formed stools is not diarrhea, nor is the passing of loose, pasty stools by breastfed babies’. Prevalence is subdivided in people below and above 5 years of age.

**Infrastructure.** WASH infrastructure is assessed using a self-developed questionnaire and spot checks (Appendix A). Both hand-washing and latrine infrastructure will be judged.

The quality of the hand-washing infrastructure is scored based on the type of hand-washing facility (i.e., running water versus bucket) (spot check), the presence of water and/or soap (spot check), and whether or not the station is always operational (questionnaire).

The quality of the latrine infrastructure is scored based on the type of facility (i.e., does it have a roof, a door, etc.) (spot check), the cleanliness (spot check), and the number of people who use the latrine (questionnaire). Information about whether or not the pit has ever been emptied, and how will also be gathered (questionnaire) as this will allow for the measurement of Sustainable Development Goal (SDG) 6 according to the WHO/UNICEF JMP (https://washdata.org) tools.

**Quality of Life.** The EQ-5D-3L is a validated questionnaire, examining 5 dimensions: Mobility, self-care, usual activities, pain/discomfort, and anxiety/depression (Appendix A). All items are scored on a 3-point Likert scale. Results of the EQ-5D-3L can be used to calculate the quality-adjusted life-year (QALY), a generic measure of disease burden. The cost per QALY will be used to determine the cost-effectiveness of each intervention arm.

**Demographics.** Demographic data will be gathered, including the Global Positioning System (GPS) location of the household, village and sub-village, name, age, gender, and education level of the household head. These demographics are crucial to verify that the correct households—i.e., the households that were randomly selected in R—are targeted at each of the intervention and data collection moments. In addition, socio-economic data is collected, i.e., age and gender of the respondent, main source of income, etc. The questionnaire is available in Appendix A.

**Compliance.** Compliance to the intervention will be monitored in real-time using KoboToolbox (https://www.kobotoolbox.org/). Monitoring will involve (1) number of household visits provided, (2) number of community sessions provided, (3) number of people who have been reached, (4) duration, (5) content of the sessions/visits, and (6) compliance of the participants. 

### 2.8. Data Analysis

Statistical analyses will be undertaken using R studio (Version 1.0.143). All analyses will be conducted on an intention-to-treat principle using all randomized participants. Demographic characteristics and baseline data will be summarized by descriptive statistics using means, standard deviations and 95% confidence intervals for continuous variables, median and inter-quartile ranges for non-normal continuous or ordinal data and percentages for categorical data. The primary and secondary outcome measures will be analyzed using generalized linear mixed models with baseline values as a covariate to assess differences between treatment groups and across time. A sensitivity analysis will be performed comparing effectiveness of the interventions in low compliance versus high compliance households. All data will be adjusted for multiple comparisons. A *p*-value < 0.05 will be considered as statistically significant.

Investigators who are involved in data analysis will be blinded to group allocation. In accordance to the European General Data Protection Regulation (GDPR) law, all private information (i.e., name or GPS) will not appear on any documents except a participant key. This key is needed to match baseline data with midline, end-line, and follow-up data. A pseudonym will be used to protect participants’ identities. The key linking the participant name to the data is not accessible to the investigators and will be destroyed after data analysis is complete.

### 2.9. Ethics and Dissemination

This study was registered at Clinicaltrials.gov (NCT03709368). An overview of the WHO registration data set can be found in Appendix A. Ethical approval was obtained at the Social and Societal Ethics committee (KU Leuven, Belgium), and at the National Institute for Medical Research (Dar es Salaam, Tanzania). In addition, a research permit was obtained at the Tanzania Commission for Science and Technology. The trial progress was overseen by both the National Institute for Medical Research and the Commission for Science and Technology. All eligible subjects will be required to provide a written informed consent, or to sign with a thumb stamp. The informed consent can be found in Appendix A. The study results will be published within 24 months of the final data collection date.

## 3. Discussion

The aim of this study is to establish an effective and feasible solution for improving hand washing behavior, latrine use, health, quality of life, and WASH infrastructure in rural LMIC. What differentiates this project from earlier work in the field is that this study examines the long-term efficacy and cost-effectiveness of a contextualized and non-contextualized add-on intervention, using a large cRCT study design and an active comparison control. 

The study objectives and protocol of this study are based on a gap analysis of the available scientific evidence and input from experts in the field [18]. The protocol was designed to involve a combination of infrastructural improvements, community meetings, and add-on household visits in order to maximize uptake by all beneficiaries. Both the contextualized and non-contextualized intervention programs endeavor to achieve resilience and lasting behavior change by implementing a multifaceted approach that relies on community- and household-based structures. Comparison with an active control cohort, involving only infrastructural improvements and community meetings, allow us to determine the added value of add-on household visits, and its cost-effectiveness.

To establish the feasibility of the program, pilot work was performed by Belgian Red Cross (BRC) in close collaboration with Tanzania Red Cross Society (TRCS). In addition, focus group discussions with locals were used to optimize all training materials. With regards to the assessments, assessors are well-trained and blinded to group allocation. Although several outcome measures are self-developed, they are based on extensive experience in the field and have gone through pilot testing. Moreover, some of the outcome measures have been aligned with WHO/UNICEF JMP tools, to allow for the measurement of Sustainable Development Goal (SDG) 6.

This thorough preparation process enables us to confidently advance into a larger cRCT to explore the efficacy of contextualized and non-contextualized interventions compared to an active training control group. After the project is finished, the findings of the study will be disseminated in rural Tanzania by means of billboards, posters, and radio spots.

## 4. Conclusions

This cRCT will allow us to validate the superiority of a contextualized versus non-contextualized versus placebo add-on interventions on improving hand washing behavior, latrine use, related health outcomes, quality of life, and hardware coverage. It will also calculate the cost-effectiveness of each approach. The knowledge that will be generated by the results of this study are likely to inform WASH research and field practice.

## Figures and Tables

**Figure 1 ijerph-16-02529-f001:**
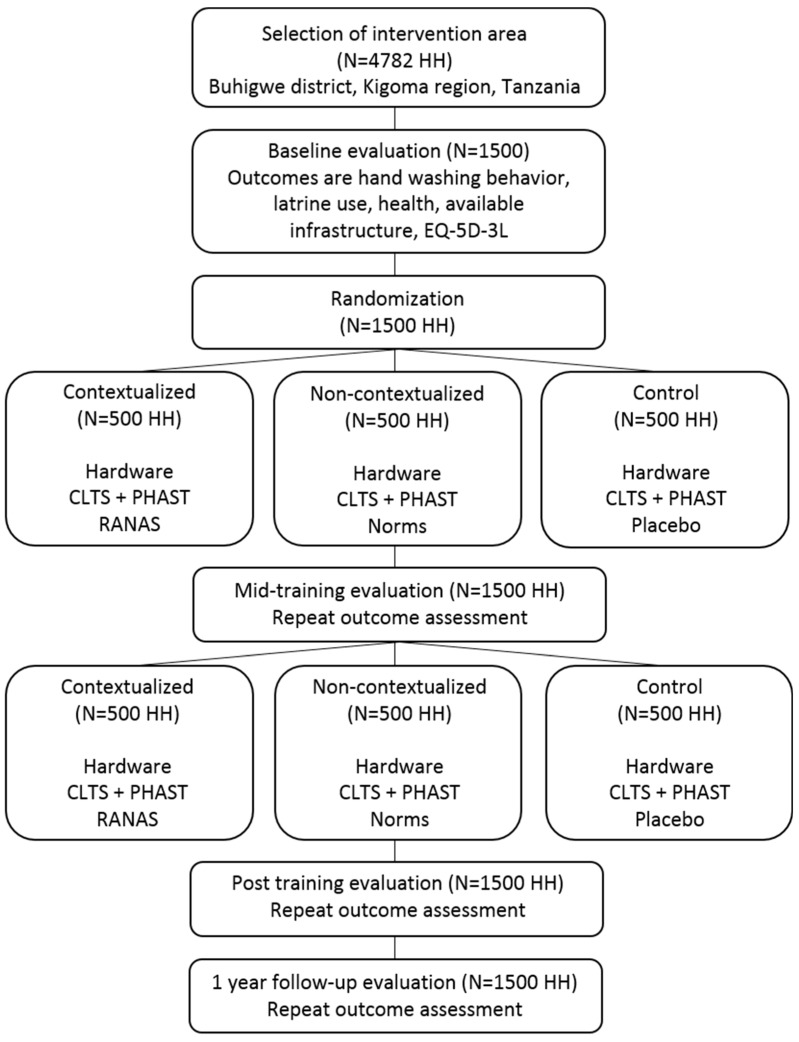
Summary of the study design and intervention protocol. HH = households; CLTS = Community-Led Total Sanitation; PHAST = Participatory Hygiene and Sanitation Transformation; RANAS = Risks Attitudes Norms Abilities Self-regulation; EQ-5D-3L = Quality-of-Life questionnaire.

**Figure 2 ijerph-16-02529-f002:**
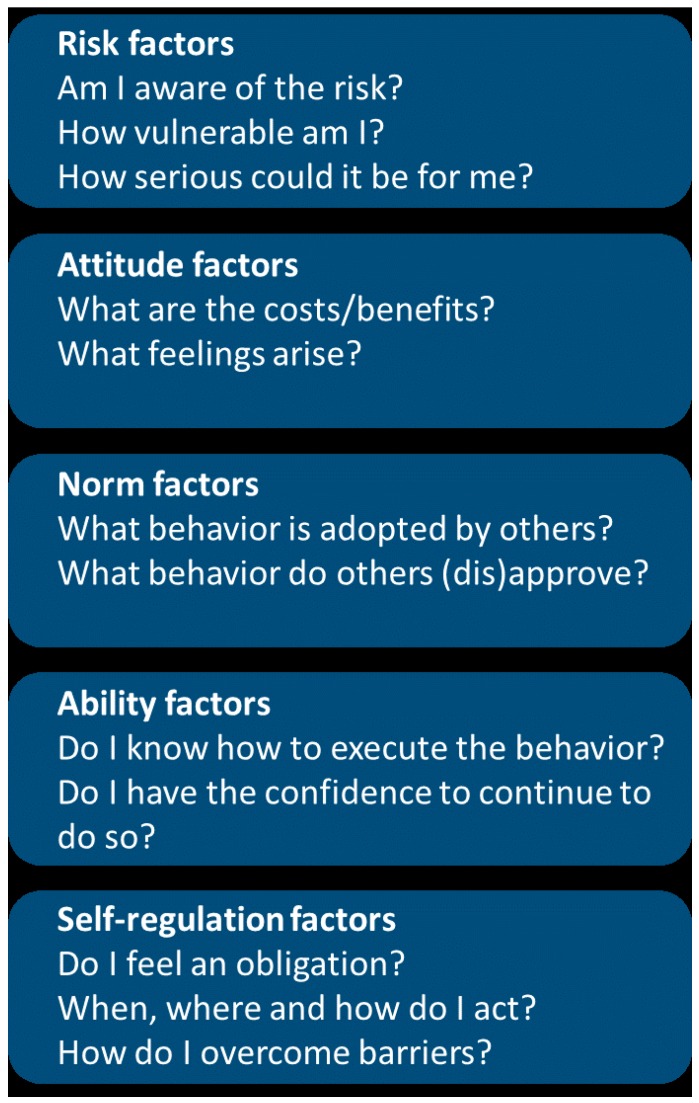
Five components of the RANAS model (https://www.ranasmosler.com/ranas).

**Table 1 ijerph-16-02529-t001:** Total population and sample in the intervention area. The intervention area consists of 3 wards, including 7 villages and 27 sub-villages. A total population of 4782 households live in the intervention area, of which a sample of 1500 households will be included in the study. Four of the sub-villages had a school.

Ward	Village	Sub-Village	Population (*n* Households)	Sample (*n* Households)
1. Nyamugali	1.1. Bulimanyi	1.1.1. Buhinda	161	50
		1.1.2. Bweru	212	67
		1.1.3. Lulengala	113	35
		1.1.4. Mudyangoti	90	28
	1.2. Nyamugali	1.2.1. Kikulazo	82	26
		1.2.2. Lukunda	108	34
		1.2.3. Mbanga	149	47
		1.2.4. Nyomvi	73	23
		1.2.5. Sakivungwe	105	33
		1.2.6. Sokoni (school)	140	44
2. Munyegera	2.1. Munyegera	2.1.1. Kabuye (school)	358	114
		2.1.2. Nyakitanga	444	140
		2.1.3. Salugale	383	118
	2.2. Songambele	2.2.1. Bilatenda	209	66
		2.2.2. Bulambila	257	82
		2.2.3. Kumsenga	234	72
		2.2.4. Nyamutukula (school)	162	51
3. Buhigwe	3.1. Buhigwe	3.1.1. Buyogwa	192	59
		3.1.2. Lugumba	204	65
	3.2. Kavomo	3.2.1. Kitagata	104	32
		3.2.2. Kitulo	116	37
		3.2.3. Mnyango	118	37
		3.2.4. Nyandela	169	54
	3.3. Mulera	3.3.1. Kamazi (school)	243	75
		3.3.2. Lukaro	116	37
		3.3.3. Muzenga	122	38
		3.3.4. Rusange	118	36

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
