# Peer review of "Effect of Contextualized Versus Non-Contextualized Interventions for Improving Hand Washing, Sanitation, and Health in Rural Tanzania: Study Design of a Cluster Randomized Controlled Trial"

_ijerph, 2019, doi:10.3390/ijerph16142529_

Round 1

Reviewer 1 Report

The analyzed manuscript describes the design and methods of a cluster randomized controlled trial that will be carried out in Tanzania. The purpose of the trial is to verify the effectiveness of contextualized and non-contextualized interventions to improve health related to hand washing and the use of latrines in different rural communities of Tanzania.

The study is very interesting and well designed and it can provide data on the efficacy of the proposed interventions in the long-term.

Author Response

Thank you for your positive feedback.

Reviewer 2 Report

Page 1, Line 42: I would suggest to add an example about the impact that adding a detergent to water could have on the efficacy of hand washing. A practical example could be:

Troiano G, Stilo A, Melcarne L, Gioffrè ME, Nante N, Messina G, et al. Hand washing in operating room: a procedural comparison. Epidemiology, Biostatistics and Public Health - Available at: http://doi.org/10.2427/11734

Page 2, Lines 21-25 and Lines 27-32: I think this part should be probably moved in the “Materials and methods” section

Page 5, Lines 1-3: I think more details should be added about the randomization process among the same subvillage (e.g how was the computer-generated random list applied in each sub-village? Were there any systemized process in place for every one of them?)

Page 5, Lines 2-3; Table 1: Although it is correctly and clearly stated that stratification was used in the randomization process, I think it should probably be mentioned in the table which sub-village has a school, and which not, or at least the total number of schools in the three arms

 Page 5, Line 4 and Page 8, Line 38: I think the Rstudio version that is going to be used should be mentioned (whether in the citation n° 29, or in the text)

Page 5, Line 14: If sanitation blocks will be built in four schools, I think it should probably be mentioned how these four schools will be distributed in the three arms

Page 7, Line 4: If the main goal of the placebo visit was to avoid jealousy in the control group, is there a reason that the clear difference against the nine additional household visits won’t raise any jealousy?

Reviewer 3 Report

This is the protocol of what will be an ambitious randomised controlled trial. Its major and inevitable weaknesses are that implementation will depend on willingness of people to rake part and that the primary outcome measure depends on their self reported behaviour. There is no obvious way of avoiding these problems and despite them, the study appears to have been funded. This is an important area for research and I hope the RCT goes well

Author Response

Thank you for your positive feedback.